



# Quantitative imaging of carbon dioxide plumes using a ground-based shortwave infrared spectral camera

Marvin Knapp[1], Ralph Kleinschek[1], Sanam N. Vardag[1,3], Felix Külheim[1], Helge Haveresch[1], Moritz Sindram[1], Tim Siegel[1], Bruno Burger[2], and André Butz[1,3,4]

[1]Institute of Environmental Physics (IUP), Heidelberg University, Heidelberg, Germany
[2]Fraunhofer Institute for Solar Energy Systems (ISE), Heidenhofstr. 2, 79110 Freiburg, Germany
[3]Heidelberg Center for the Environment (HCE), Heidelberg University, Heidelberg, Germany
[4]Interdisciplinary Center for Scientific Computing (IWR), Heidelberg University, Heidelberg, Germany

**Correspondence:** Marvin Knapp (marvin.knapp@uni-heidelberg.de)

**Abstract.** We present the first results of a ground-based imaging experiment using a shortwave infrared spectral camera to quantify carbon dioxide ($CO_2$) emissions from a coal-fired power plant in Mannheim, Germany. The power plant emits more than $4.9 \, \mathrm{MtCO_2/year}$ and is a validation opportunity for the emission estimation technique. The camera is a hyperspectral imaging spectrometer that covers the spectral range from $900 \, \mathrm{nm}$ to $2500 \, \mathrm{nm}$ with a spectral resolution of $7 \, \mathrm{nm}$. We identify $CO_2$ enhancements from hourly averaged images using an iterative matched filter retrieval using the $2000 \, \mathrm{nm}$ absorption band of $CO_2$. We present 11 plume images from five days in 2021 and 2022 covering a variety of ambient conditions. We design a forward model based on a three-dimensional, bent-over Gaussian plume rise simulation and compare our observed emission plumes with the forward model. The model depends on the parameters ambient wind velocity, wind direction, plume dispersion, and emission rate. We retrieve the emission rate by minimizing the least-squares difference between the measured and the simulated images. We find an overall reasonable agreement between the retrieved and expected emissions for power plant emission rates between $223 \, \mathrm{tCO_2/h}$ and $587 \, \mathrm{tCO_2/h}$. The retrieved emissions average to 89% of the expected emissions and have a mean relative uncertainty of 25%. The technique works at wind speeds down to $1.4 \, \mathrm{m/s}$ and can follow diurnal emission dynamics. We also include observations with unfavorable ambient conditions, such as background heterogeneity and slant observation angles. These conditions are shown to produce considerable biases in the retrieved emission rates, yet they can be filtered out reliably in most cases. Thus, this emission estimation technique is a promising tool for independently verifying reported emissions from large point sources and provides complementary information to existing monitoring techniques.

## 1 Introduction

Carbon dioxide ($CO_2$) is the most important anthropogenic greenhouse gas driving climate change (Masson-Delmotte et al., 2021). The signatory countries of the Paris Agreement set ambitious goals for $CO_2$ emissions reductions, and they agreed on implementing a stock taking mechanism to supervise mitigation progress. In consequence, the international science community has elaborated a plan to build a monitoring and verification support (MVS) capacity for anthropogenic $CO_2$ emissions (Janssens-Maenhout et al., 2020). Besides refining inventory-making and atmospheric modeling, the plan includes further de-



veloping and implementing atmospheric measurement techniques that can help quantify anthropogenic emissions from global to local scales using in-situ and remote sensing observations with the goal to independently verify reported emissions and to
monitor the effectiveness of reduction measures.

The local scales have received particular attention recently since spectroscopic imaging techniques are emerging that enable quantification of localized anthropogenic emission sources. Pioneering space missions such as the Greenhouse Gases Observing Satellite (GOSAT) and the Orbiting Carbon Observatory (OCO-2/-3) were able to quantify the urban $CO_2$ domes of megacities such as Los Angeles (Kort et al., 2012; Schwandner et al., 2017; Kiel et al., 2021). OCO-2 was the first satellite to
deliver images of $CO_2$ plumes from individual coal-fired power plants (Nassar et al., 2017), and since then, manifold activities have demonstrated that spectroscopic imaging with fine spatial resolution from satellites and aircraft can deliver facility-scale emission estimates for $CO_2$ (Thorpe et al., 2017; Cusworth et al., 2021a; Fujinawa et al., 2021) and, using analogue techniques, for methane ($CH_4$) (Frankenberg et al., 2016; Duren et al., 2019; Guanter et al., 2021). Emission estimation methods either use a mass-balance approach (Liu et al., 2021; Varon et al., 2018, 2020), Gaussian plume models (Nassar et al., 2017; Schwandner
et al., 2017; Varon et al., 2018), or machine learning (Jongaramrungruang et al., 2022). Many activities aim at designing next generation satellite missions with ground resolution on the order of a few ten meters and enhanced quantification capabilities for $CO_2$ and $CH_4$ (Strandgren et al., 2020; Jacob et al., 2022). On the ground, pilot studies and field campaigns were able to constrain city-scale and localized emissions sources using various techniques (Christen, 2014) ranging from eddy-covariance methods (Crawford and Christen, 2015; Christen et al., 2011) to local in-situ and remote sensing concentration measurements
deployed on mobile platforms and in small ad-hoc networks (Hase et al., 2015; Luther et al., 2022). Gålfalk et al. (2016) report on the development of a $CH_4$ camera that operates in the thermal infrared, enabling imaging of $CH_4$ released from localized sources. Our precursor study (Knapp et al., 2023) demonstrates the methane imaging capabilities of a stationary spectral camera in the shortwave infrared (SWIR) and derives sub-hourly emission estimates for coal mine ventilation shafts.

Here, we present the first results from our ground-based SWIR camera (HySpex SWIR-384) on imaging of $CO_2$ plumes from
strong point sources. The camera collects sunlight scattered in the sky, and the retrieval exploits the $CO_2$ absorption structures around 2000 nm wavelength, which is analogue to previously reported aircraft and satellite techniques that work on reflected sunlight (Thorpe et al., 2017). Thus, the measurements only work in daytime and require fair weather conditions. Observing skylight in the SWIR further comes with the disadvantage that the sky, while being spectrally smooth, is dark, and thus, our technique currently requires co-adding on hourly timescales. This, however, might be improved by future developments in
sensor technology. We develop an emission estimation technique based on bent-over Gaussian plume modeling and use the well-known emissions of a hard-coal power plant to validate our results. Such ground-based $CO_2$ cameras could be particular useful to monitor emissions of point-like sources such as coal-fired power plants, cement factories, industrial facilities, or natural sources such as volcanoes. These observations would be a valuable addition to our MVS capacities per se and could be used together with network techniques to disentangle contributions from various sources in complex emission landscapes. We
also suggest that imaging $CO_2$ plumes might be a tool to raise public awareness on the scale of local greenhouse gas pollution and the urgency to implement emission reductions (Jungmann et al., 2022).





For demonstration purposes, we deploy the HySpex hyperspectral camera together with a portable wind lidar at a few kilometers distance from a medium-sized hard-coal power plant (>4.9 $MtCO_2/yr$) located in Mannheim, Germany (section 2 for measurement setup). Repeatedly scanning the sky above the power plant, we retrieve $CO_2$ column enhancements in the power plant plume using a matched filter algorithm similar to previous satellite (Guanter et al., 2021) and airborne studies (Foote et al., 2020, 2021) (section 3.2 for data analysis). Based on the plume enhancements, we estimate the emissions of the power plant from the observed plumes. To this end, we design a forward model to match our observations based on the plume rise model IBJpluris (Janicke and Janicke, 2001). The best match between the measured and simulated observations feeds our emission estimate (section 4 for the emission estimation method). In total, we collect 11 plumes over 5 days, for which we compare our emission estimates to the emissions of the power plant calculated from the instantaneous power production. Finally, we discuss the capabilities and limitations of the technique (section 5 for the results).

## 2 Instrumentation and field-deployment

### 2.1 HySpex SWIR-384 camera

The HySpex SWIR-384 camera is a commercially available hyperspectral camera by Norsk Elektro Optikk® (NEO). The camera optics focuses the incoming light onto a slit, which passes light from a horizontal opening angle of 7.3 mrad width. After collimation, a grating disperses the incoming light, and a mirror focuses the spectrum on a 2D-detector array. The detector array samples the vertical dimension of the scene with 384 pixels (384 "lines") and the spectral dimension with 288 pixels ("288 channels"). The lines of the detector cover a vertical field of view of 16° and each pixel covers a solid angle of approximately 7.3 mrad × 7.3 mrad. The channels record the spectrum between 950 nm and 2500 nm with a sampling of 5.45 nm. Each read-out of the 2D-detector is called a frame. The camera is mounted on a rotation stage, which turns it in a horizontal (azimuth) direction, thus scanning over the target scene by collecting a sequence of frames. The rotation pattern is clockwise in steps that correspond to the acceptance angle of a single pixel. A shutter closes the camera aperture prior to and after each scene scan in order to take 200 dark spectra. The detector is cooled to 147 K during operation to reduce dark current. We refer to taking a hyperspectral image as a scan. The raw output is in analog-digital-converter units with a 16-bit resolution, representing radiances between 1 and $2^{16} - 1$, and the datacube has dimensions of number of frames × number of lines × number of channels. A tripod carries the rotation stage with the camera during field deployment. A rugged, field-deployable GETAC® laptop controls the camera and the rotation stage. The camera and the laptop each run on their own battery packs, which provide enough power for more than six hours of consecutive measurements. Figure 1 shows the camera deployed in the field on 2021/09/06 observing the sky above the power plant in Mannheim.

The HySpex camera comes with a radiometric and spectral calibration by the manufacturer. For calibrating the instrument spectral response function (ISRF), we used a tunable diode laser operating around 1600 nm in our lab, similar to the setup of Lenhard et al. (2015). The ISRF has a Gaussian-like profile with a FWHM of 6.7 - 7 nm, see figure 2. This is in agreement with the manufacturer calibration results of approximately 7 nm for the 1694 nm Argon emission line. We use a radiative transfer



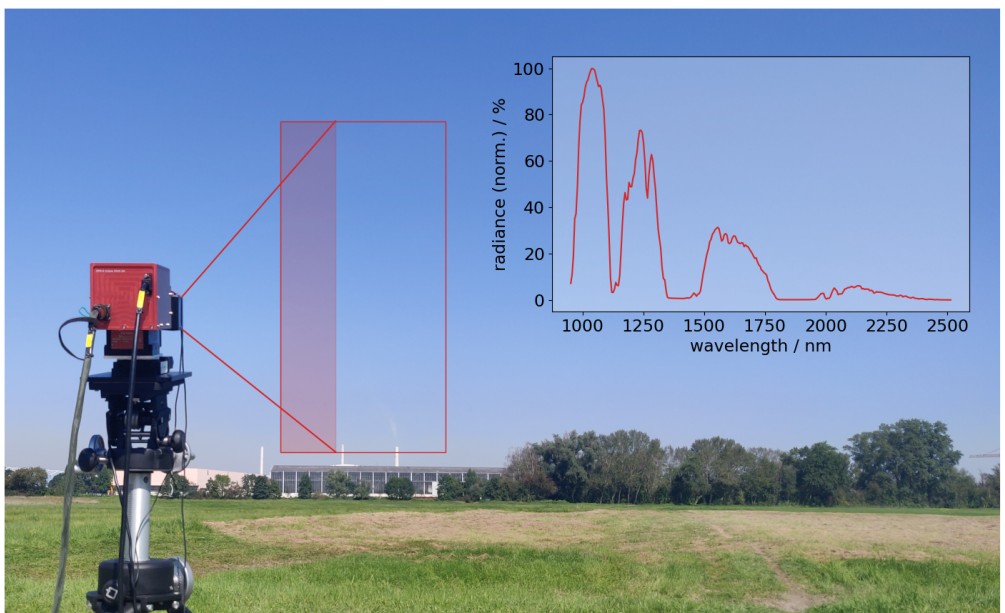

**Figure 1.** Photograph of the HySpex SWIR-384 camera deployed in the vicinity of the hard-coal power plant Grosskraftwerk Mannheim on 2021/09/06. In this particular case, the field of view (red frame) covers the sky above two stacks of the power plant. The instrument scans the scene from left to right. The inset panel shows a typical sky spectrum in the entire accepted spectral range from roughly 900 to 2500 nm. Notice the absorption features of carbon dioxide at around 2000 nm.

model similar to Guanter et al. (2021) for the spectral calibration of our fit interval and correct an offset of +2.25 nm to the manufacturer's calibration. Figure 1 shows a typical clear sky spectrum with the entire recorded spectral range.

## 2.2 Attitude and Heading Reference System

We constrain the observation geometry during measurements with an Attitude and Heading Reference System (AHRS). An MTi-7 Miniature GNSS/INS Module from XSENS® was mounted on the HySpex camera such that it rotates with the camera during observation. The device performed well in previous campaigns in ship-borne applications (Dörner et al., 2018). We conducted performance tests on its inertial navigation system (INS) and found its 10 min precision well below 0.05° for the instrument pitch and roll angle. All data transfers in real time to the GETAC® laptop controlling the HySpex camera. The sensor does not provide reliable data on the instrument viewing azimuth angle (VAA) in stationary operation. We find the VAA of each image frame by identification of a distinct landmark within the image, e.g., a chimney, and assigning the forward azimuth angle from the camera location to the landmark. The VAA of all other frames in the image follows from the horizontal opening angle of the camera. For typical distances between the camera and the landmark, the uncertainty of the VAA is below 1°. Additionally, the sensor provides us with GPS/GNSS-based geolocation data. We use the software of Holmgren et al. (2018) to calculate the solar zenith angle (SZA) and solar azimuth angle (SAA) from the geolocation and time of the observation.





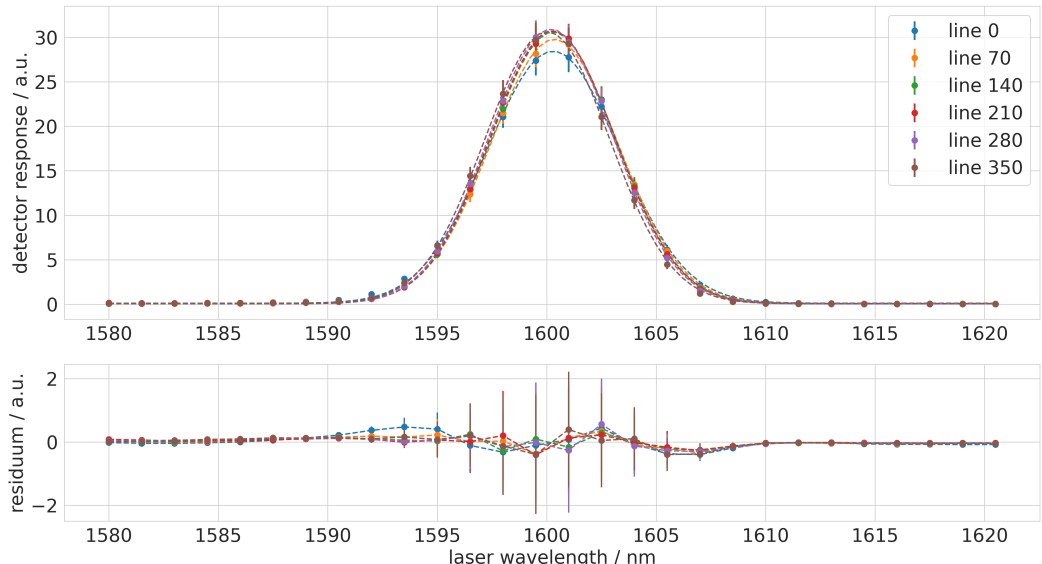

**Figure 2.** The instrument spectral response function of channel 119 at selected lines across the detector. The upper panel shows the measured radiance and a fitted Gaussian. Each data point represents a mean and standard deviation of 286 frames. The lower panel shows the residuum of the data points to the Gaussian fit.

## 2.3 Windranger 200 lidar

The Windranger 200 is a compact and lightweight wind lidar produced by METEK®. Adler et al. (2021) used it to successfully
study boundary layer turbulence. The instrument weighs approximately 50 kg and has dimensions of 840 mm x 540 mm x
580 mm. For absolute reference of wind direction, the Lidar needs to point to the North or another known azimuth reference
point. The Lidar works with the Frequency Modulated Continuous Wave (FMCW) technology (Peters, 2018). It uses the
Doppler-shifted back-reflection of a 1545 nm laser for the measurements. The laser beam rotates by 360° once per second,
producing a data point of wind velocity and direction at one height layer. It measures at six height levels up to 200 m. In our
setting, the layers are at 10 m, 20 m, 50 m, 100 m, 150 m, and 200 m. The method is sensitive to turbulent wind fluctuations
within one rotation, but it provides a complete profile approximately every 10 seconds, such that statistical fluctuations can be
reduced by averaging over several rotations.

## 2.4 Field measurements

The HySpex operates in a 3 km to 4 km distance from the target, thus an image has a spatial resolution of approximately 2 m
to 3 m. This provides us with a spatially well resolved plume image, while we measure enough background sky pixels for
the retrieval of enhancements (section 3.2). We level the camera based on the INS real-time data prior to each observation
by collecting 10 min means of the INS pitch and roll and adjusting the camera's alignment for potential offsets (section 2.2).



**Table 1.** The first four columns list the date, the time interval of the observations, the unit number of the power plant that was observed, and if the Windranger 200 Lidar was available. Columns 5 and 6 list the mean viewing azimuth angle (VAA) of the camera and the distance $d$ to the observed unit. Columns 7 and 8 list the AERONET aerosol optical thickness (AOT) at $2000\,\mathrm{nm}$ and the asymmetry parameter ($g$) of the scattering phase function, see section 3.3. Column 9 lists if condensation was observed in the plume. For more detailed atmospheric conditions, see table S2.

| Date | Time [UTC] | Unit | Lidar | VAA [°] | d [m] | AOT | $g$ | condensation |
|------|-----------|------|-------|---------|-------|-----|-----|--------------|
| 2021/09/08 | 12:13 - 16:36 | 9 | no | 347 (north) | 3183 | 0.012 | 0.82 | no |
| 2022/03/23 | 14:51 - 17:36 | 9 | yes | 94 (east) | 3760 | 0.022 | 0.78 | yes |
| 2022/03/26 | 13:31 - 17:36 | 9 | yes | 347 (north) | 3179 | 0.035 | 0.77 | yes |
| 2022/03/28 | 15:35 - 16:28 | 9 | yes | 91 (east) | 3760 | 0.063 | 0.76 | yes |
| 2022/05/13 | 12:21 - 15:39 | 6 | yes | 333 (north) | 4098 | 0.065 | 0.76 | no |

This ensures a roll angle of $0.0\pm0.1°$. Subsequently, the Viewing Elevation Angle (VEA) is adjusted by tilting the camera such that the lower edge of the image contains the upper part of the $CO_2$ source, e.g., the chimney tip, and the upper part of
120 the image shows the sky (figure 1). Typical single exposures of the detector range between $8\,\mathrm{ms}$ and $20\,\mathrm{ms}$. Each frame adds 5 - 10 of these exposures before the camera rotates a step in the azimuth direction. Scans that cover azimuth angles of $10°$ to $15°$ have typical scan times of $90\,\mathrm{s}$. We record consecutive scans as long as atmospheric conditions like cloud cover and solar illumination are favorable, i.e., a bright and clear sky. To achieve sufficient signal-to-noise, the hyperspectral datacubes submitted to the retrieval are co-additions of scans of at least 55 minutes.

125 The LIDAR is positioned next to the camera. Its power supply is a Jackery® Explorer 500 battery with a Jackery® SolarSaga 100W solar panel. During field observations, we monitor the wind data quality and re-calibrate the Lidar height levels if data quality decreases. We measure wind velocities and directions in six levels from $10\,\mathrm{m}$ to $200\,\mathrm{m}$ height.

Here, we present observations taken on five days in 2021 and 2022. We headed to the field when weather forecasts predicted clear skies and targeted the largest section of the power plant, unit 9, preferentially. Table 1 lists the observation periods along-
130 side a short description of atmospheric conditions and the instrumental setup. Atmospheric conditions varied over the day and between days concerning cloud cover, wind conditions, and aerosol load. For more detailed information on the atmospheric conditions, see section 4.3 and table S2. We process scans that pass a visual quality filter for heavy clouds or sporadic obstruction by pedestrians or cars. Time intervals with favorable conditions vary between 100 and 258 minutes per day, and we retrieve plume images for averaged images of approximately one hour, as described in section 3.1.

135 We observed unit 9 on all days except 2022/05/13, on which unit 9 malfunctioned, and we observed unit 6 instead.



## 3 Carbon dioxide retrieval

### 3.1 Preprocessing the images

For each scan, we term the raw detector output of the camera $DN_{ijk}$ where $i$ labels the channel, $j$ the line, and $k$ the frame. We take 200 background frames prior to and after each scan. Background frames are taken with a closed shutter in front of the detector. We calculate a mean detector background $BG_{ij}$ before and after the scan as the average of the background frames. The backgrounds are linearly interpolated to the frames $k$ during observation for each detector element and subtracted from each frame. The background corrected image $L_{ijk}$ is therefore given by

$$L_{ijk} = DN_{ijk} - \frac{BG_{ij}^{after} - BG_{ij}^{before}}{\text{number of frames}} \cdot k. \tag{1}$$

For later convenience, we chain the spectral dimension $i$ into a vector $\boldsymbol{L}_{jk} = [L_{1jk}, L_{2jk} \cdots L_{288jk}]$.

Malfunctioning detector elements (pixels on the detector) are identified in laboratory measurements using halogen lamps fed into an integrating sphere (40 cm diameter). Elements are removed from further processing if they exhibit an exceptionally high, low, or variable response to the broadband illumination. The missing values are interpolated in spatial (line) direction. Field deployment of the HySpex camera typically produces several hundred single scans. Each scan is corrected according to eq. (1) and visually inspected for quality control. We remove images that are corrupted by obstacles obscuring the field of view in single frames, or by significant changes in the overall atmospheric conditions, e.g., cloud formation. The scans remaining after quality control are averaged. Some observed emission plumes show condensation from the water vapor co-emitted with the $CO_2$. The light paths of photons under clear sky is substantially different from those under condensate conditions. To avoid intermingling light path differences with concentration differences, we identify and exclude pixels with condensate from the averaged image. The saturation of each pixel in a scan is defined as the maximum of the spectrum in the pixel $S_{jk} = \max\limits_{i}(L_{ijk})$. We calculate the background saturation $BS_{jk}$ of the sky for each scan by

$$BS_{jk} = \underset{j}{\text{med}}(S_{jk}) \cdot \left( \frac{\underset{k}{\text{med}}(S_{jk})}{\left\langle \underset{k}{\text{med}}(S_{jk}) \right\rangle_{j}} \right)^{T}, \tag{2}$$

where $\text{med}$ is the median operator. Equation (2) calculates the clear sky saturation for each pixel in a scan from the scan itself, assuming that the plume condensate covers a small part of the sky. We subtract $BS_{jk}$ from the scan saturation, such that the residual image scatters around zero where a clear sky was observed. The use of the median assures that the high reflectivity of the condensed water does not affect $BS_{jk}$. Pixels are masked in scans if their saturation deviates more than $+3\sigma$ from the residual distribution since this shows an exceptionally high reflectivity caused by plume condensation or clouds. We build the averaged image from the spectra of the single scans, which were not removed by the saturation mask. Therefore, the averaged image contains pixels to which not all scans have contributed. We exclude pixels from the retrieval of the averaged image if



less than 90 % of the scans contribute to the average image. Figure 3 shows how the mask based on equation (2) identifies

over-saturated pixels in single scans and how the averaged image is constructed from there.

We use the geolocation data of the camera and the target to calculate distance $d$ and the viewing azimuth angle of the instrument based on the WGS84 reference system (Slater and Malys, 1998). The observed chimney also provides us with a line of known height $h$ in the image, thus the viewing elevation angle is found by

$$\text{VEA} = \arctan\left(\frac{h}{d}\right). \tag{3}$$

The geometric area of each image pixel $A_j$ at the distance $d$ to the target source is

$$A_j = (d \cdot \tan(\Delta\text{VEA}_j)) \cdot (d \cdot \tan(\Delta\text{VAA})), \tag{4}$$

where $\Delta\text{VAA} = 0.73\,\text{mrad}$ is the horizontal opening angle of the camera and $\Delta\text{VEA}_j \approx 0.73\,\text{mrad}$ is the vertical opening angle of each line.

Inhomogeneities in detector response or the optical setup cause striping patterns in imaging spectrometer data (Borsdorff

et al., 2019). We compensate for this by dividing the spectral vectors $\boldsymbol{L}_{jk}$ element-wise by a reference spectrum $\hat{\boldsymbol{L}}_j$ from the same line. Thus, our normalized spectral vectors $\boldsymbol{l}_{jk}$ are given by

$$\boldsymbol{l_{jk}} = \frac{\boldsymbol{L}_{jk}}{\hat{\boldsymbol{L}}_j}, \tag{5}$$

and scatter around unity if the scene was homogeneous. The reference spectrum is taken from each scan itself as the mean of 20 frames upwind of the source.

**3.2 Matched filter retrieval**

The matched filter is a data-driven statistical approach for signal identification and quantification in noisy data, e.g., hyperspectral images (Dennison et al., 2013; Manolakis et al., 2014; Zhang et al., 2020; Cusworth et al., 2021b; Guanter et al., 2021). It estimates a spectral background variability from two-dimensional spatial tiles of absorption spectra and identifies a pre-defined spectral signature exceeding this variability, i.e., in our case, the spectral absorption signature of $CO_2$ around 2000 nm wave-

length. Foote et al. (2020) improve on the classical matched filter by introducing an albedo correction and a sparsity constraint on the enhancements, which we also used in our precursor study for $CH_4$ (Knapp et al., 2023). This version of the matched filter iteratively removes the target signal from the background clutter, which is particularly important for the detection of weak signals in the presence of strong background signals (Foote et al., 2020; Schaum, 2021; Pei et al., 2023). Heterogeneities in the observed scene are particularly challenging for statistical retrievals (Ayasse et al., 2018). Thus, the homogeneous reflectivity

of the clear sky is an advantage for the matched filter, although the sky is rather dark in the shortwave infrared spectral range.

The classic matched filter (CMF) calculates the enhancement $\alpha$ in each pixel as





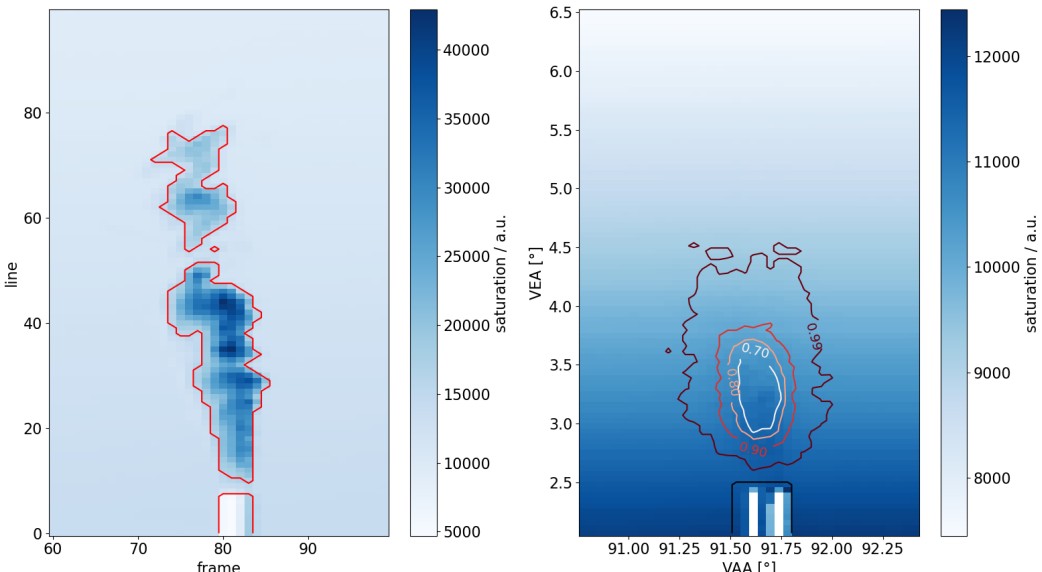

**Figure 3.** (Left panel) Saturation of a single scan focused on the part of the image with a condensate plume. The image was taken on 2022/03/23 in Mannheim. The red border surrounds all pixels that are excluded from averaging. (Right panel) Average image with colored lines marking the areas in which a certain fraction of scans contributed to the spectra in the pixels. The black line marks a mask (generated by eye), which excludes the chimney from further processing. Note that the image dimensions are VAA and VEA.

$$\alpha_{jk} = \frac{(l_{jk} - \mu)^T \mathbb{C}^{-1} t}{t^T \mathbb{C}^{-1} t}, \tag{6}$$

where $l_{jk}$ are the normalized spectral vectors defined in equation (5), $\mu$ is the mean normalized spectral vector across the image, and $\mathbb{C}$ is the covariance matrix of all the normalized spectral vectors across the image. The target signature $t$ contains the spectral feature the matched filter retrieval is sensitive to. In our case, it follows from the absorption cross-section of $CO_2$. The target signature is commonly derived from the assumption that an enhancement $\alpha$ in a pixel acts on a spectrum $L$ according to Beer-Lambert's law

$$L = L_0 \cdot \exp(-\alpha \cdot s), \tag{7}$$

where $s$ is the unit absorption spectrum of the gas (section 3.3). We rearrange equation (7) for referenced spectra and linearize the exponential function via Taylor expansion to first order, which yields

$$\frac{L}{L_0} \approx 1 - \alpha \cdot s. \tag{8}$$



The target signature $t$ of the matched filter is given by the derivative of equation (8) with respect to $\alpha$, which is the unit absorption spectrum $s$.

We adapt our retrieval from the MAG1C (Matched filter with Albedo correction and reweiGhted L1 sparsity Code) algorithm
of Foote et al. (2020), which decreases enhancement uncertainty and background noise. The albedo correction reduces the effect of background heterogeneity, which accounts for the gradient in the brightness of the clear sky in our case. We calculate a scalar factor

$$r_{jk} = \frac{\boldsymbol{L}_{jk}^T \hat{\boldsymbol{L}}_j}{\hat{\boldsymbol{L}}_j^T \hat{\boldsymbol{L}}_j} \tag{9}$$

for each pixel, which gives its brightness relative to the line reference spectrum. The sparsity prior adds an L0-regularization
to the cost function of the retrieval to minimize the number of detected enhancements since we expect enhancements in less than 1‰ of the pixels. Foote et al. (2020) further introduce a positivity-constraint on $\alpha$ since gas enhancements in the atmosphere are non-negative. Thus, equation (6) transforms to an iterative retrieval given by

$$\alpha_{jkn} = \max\left(\frac{(\boldsymbol{l}_{jk} - \boldsymbol{\mu}_n)^T \mathbb{C}_n^{-1} \boldsymbol{s}_j - w_n}{r_{jk}\boldsymbol{s}_j^T \mathbb{C}_n^{-1} \boldsymbol{s}_j}, 0\right), \qquad\qquad w_n = \frac{1}{\alpha_{n-1} + \kappa}, \tag{10}$$

where $w_n$ is the pixel's regularization weight of the sparsity prior, $n$ the iteration step, and $\kappa$ a small number for numerical
robustness. The target signature is given by the unit absorption spectrum $\boldsymbol{s}_j$ of the gas, which depends on the image line $j$ (section 3.3). The iteration is necessary due to the sparsity constraint (Candès et al., 2008). It allows for iteratively improving on the estimate of the background mean and covariance, $\boldsymbol{\mu}$ and $\mathbb{C}$. We perform 30 iteration steps for each scene to get an accurate estimate of the background distribution. Pixels with low abundances are artificially forced to zero in the MAG1C algorithm by Foote et al. (2020). To avoid a systematic bias from these pixels in the emission inversion (section 4.4), we perform the
final iterative retrieval step without the positivity and sparsity constraint from equation 10. This ensures that the background estimation ($\boldsymbol{\mu}, \mathbb{C}$) benefits from the iterative retrieval, while the retrieved enhancements are non-zero in the background pixels of the image. Thus, we obtain a final estimate of the enhancement $\alpha_{jk}$ in every pixel. The uncertainty of the matched filter retrieval is given by the retrieval error covariance matrix (Köhler et al., 2015) and depends on $s$ and $\mathbb{C}$. It evaluates to

$$\sigma_j^2 = \frac{1}{\boldsymbol{s}_j^T \mathbb{C}^{-1} \boldsymbol{s}_j}, \tag{11}$$

which gives an enhancement variance $\sigma_j^2$ for each line of the image.

### 3.3 Unit absorption spectrum

In our units, the unit absorption spectrum $s$ defined in equation 8 is the relative change of observed radiance due to a $1\,\mathrm{ppm}$ increase in the atmospheric mixing ratio of $CO_2$ over a path length of $1\,\mathrm{m}$. The target spectrum can be calculated from the



absorbing molecules' absorption cross-section by radiative transfer calculations and convolution with the instrument function

(Thorpe et al., 2013; Thompson et al., 2015). Foote et al. (2021) show that a scene-specific spectral signature greatly improves the quality of the retrieved data. Thus, we choose to simulate the change of radiance with the atmospheric enhancement of $CO_2$ for a ground-based observer. An enhancement $\alpha$ of the atmospheric column of an absorber increases the optical thickness such that the at sensor radiance (ASR) $I$ changes according to Beer's law

$$I(\lambda; \alpha) = I_0(\lambda) e^{-\tau(\alpha)}, \tag{12}$$

where $\tau$ is the optical thickness due to $\alpha$ and $I_0$ the observed radiance of an atmosphere without an enhancement. This assumes that there is no change in the photon light path induced by the absorber, or a co-emitted substance like water vapor. To simulate the effect of an additional column of carbon dioxide on the spectrum, we calculate the optical thickness $\tau$ as

$$\tau = n_{CO_2} \cdot \sigma_{CO_2} \cdot \Delta z \tag{13}$$
$$= n_{air} \cdot \sigma_{CO_2} \cdot \alpha \cdot 10^{-4}, \tag{14}$$

where $n_{CO_2}$ and $n_{air}$ are the number densities of carbon dioxide and ambient air in $\frac{molecules}{cm^3}$, respectively, $\sigma_{CO_2}$ is the absorption cross-section of carbon dioxide in $\frac{cm^2}{molecule}$, $\Delta z$ the path length in cm, $\alpha$ the enhancement in $ppm \cdot m$, and $10^{-4}$ is a unit conversion factor from $cm^2$ to $m^2$. Both, $I_0(\lambda)$ and $I(\lambda; \alpha)$, assume infinite spectral resolution of the instrument i.e., they are line-by-line quantities. Thus, we need to convolve the ASR with the ISRF of the instrument to get simulated measurements $F(\lambda; \alpha)$,

$$F(\lambda; \alpha) = \int I(\lambda'; \alpha) \cdot \mathcal{N}(\lambda - \lambda'; FWHM) \, d\lambda', \tag{15}$$

where $\mathcal{N}$ is the Gaussian kernel with an FWHM of 7 nm. From equation (7), it follows that

$$s(\lambda) = -\frac{\partial}{\partial \alpha} \ln\left(\frac{F(\lambda; \alpha)}{F_0(\lambda)}\right), \tag{16}$$

where $s$ is the unit absorption spectrum in units of $\frac{1}{ppm \cdot m}$. It is binned to each channel $i$ according to the spectral calibration of the instrument.

We calculate the ASR using the single scattering approximation of the radiative transfer equation (RTE) for an upward looking observer. Since we are only interested in a relative change of radiance, we simplify the RTE by neglecting thermal emission and multiple scattering. We consider scattering on molecules using Rayleigh theory and scattering on aerosols via a Henyey-Greenstein phase function (Henyey and Greenstein, 1941). These assumptions lead to an analytically solvable RTE of the form



$$I(\lambda) = S_0(\lambda) \cdot \frac{\gamma_0}{4\pi(|\gamma| - |\gamma_0|)} \cdot \sum_k^{layer} \tilde{\omega}_k(\lambda) p_k(\lambda; \Omega, \Omega_0) e^{-\frac{\tau_k(\lambda)}{|\gamma_0| - |\gamma|}} \left( 1 - e^{-\Delta\tau_k \left( \frac{1}{|\gamma_0|} - \frac{1}{|\gamma|} \right)} \right),$$ (17)

where $S_0$ is the top of the atmosphere (TOA) radiance, $\gamma_0 = \frac{1}{\cos(\text{SZA})}$, $\gamma = \frac{1}{\cos(\text{VZA})}$, $\tilde{\omega}$ is the single scattering albedo, and $p$ the scattering phase function for the beam direction before ($\Omega_0$) and after ($\Omega$) the single scattering. Equation (17) holds for an atmosphere consisting of plane-parallel, horizontally homogeneous layers with an optical thickness $\Delta\tau_k$ per layer $k$ and a total atmospheric optical thickness down to a layer k of $\tau_k$.

We solve equation (17) between 1900 nm and 2300 nm at a resolution of 0.001 nm to calculate $I_0(\lambda)$. The model uses absorption cross-sections of $CO_2$, $CH_4$, and water vapor from the HITRAN2016 database (Gordon et al., 2017) and the US standard atmosphere from Anderson et al. (1986), in which the mean background concentrations of $CO_2$ and $CH_4$ are scaled to 420 ppm and 1.81 ppm, respectively. The simulation of the radiance vector builds on 100 layers of 400 m thickness. Since we observe in slant viewing elevation angles, we use an empirical adjustment by Kasten and Young (1989) for SZAs above 70°
and VEAs below 20° to correct for the spericity of the atmosphere.

   Foote et al. (2021) show the importance of scene-specific unit enhancement spectra in a matched filter retrieval for significant parameters of airborne instruments, i.e., the atmospheric water column, surface elevation, solar zenith angle, and sensor altitude. We adapt their technique and compute the unit absorption spectra specifically for each observation period. Specific quantities are the mean SAA and VAA during observation as well as the aerosol optical thickness and scattering phase function. We assign
an SZA and SAA to each observation from astronomical calculations based on the observation location and time and VEA and VAA based on geometrical calculations (section 3.1). Aerosol information is taken from the closest AERONET station in Karlsruhe, Germany. We use the daily mean of the aerosol optical thickness (AOT) at 870 nm and the Angström exponent (AE) to calculate the AOT at 2000 nm. Aerosol concentrations are assumed to be horizontally homogeneous and vertically decreasing exponentially with the atmospheric scale height. We use the asymmetry parameter of AERONET at 1020 nm,
which is the largest wavelength available, to compute the Henyey-Greenstein phase function. Furthermore, we compute a look-up table of unit absorption spectra for a set of viewing geometries, using a specific unit absorption spectrum for each line in an image since the VEA covers a 16° range. Thus, our unit absorption spectrum is specific to the changing geometry within a single observation. The look-up table contains unit absorption spectra for 5 SZAs (10°, 30°, 50°, 70°) and 7 VEAs (2°, 5°, 8°, 11°, 14°, 17°, 20°). We interpolate a specific unit absorption spectrum for each line from the look-up table with the image
mean SZA and the lines' VEA. Figure 4 shows $CO_2$ unit enhancement spectra at an SZA of 50° (right panel) at an VEA of 8°. Both plots show the same expected behavior from the observation geometry. The longer the light path in the atmosphere, the more light is absorbed by background $CO_2$. Thus, an additional enhancement has a smaller effect on the ASR, and the absolute values of the unit absorption spectrum decrease.



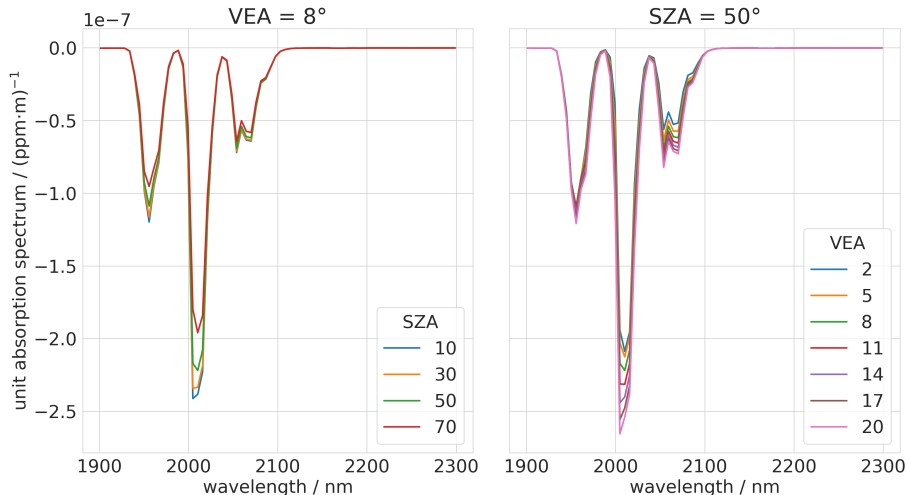

**Figure 4.** $CO_2$ unit absorption spectra for the HySpex SWIR-384 camera. The left panel shows the dependency on the solar zenith angle (SZA), the right panel the dependency on the viewing elevation angle (VEA).

## 4 Estimating emission rates

Various methods have been developed to estimate emission rates from images of $CO_2$ and $CH_4$ enhancement plumes of localized sources. $CH_4$ has received more attention recently, but the methods work mostly analogously for both gases. Gaussian plume models (GPMs) are widely used to estimate emission rates by fitting the model to snapshot observations of top-down observed plume images (Bovensmann et al., 2010; Krings et al., 2011; Rayner et al., 2014; Matheou and Bowman, 2016; Nassar et al., 2017; Schwandner et al., 2017). GPMs simulate the plume spread from point sources along a horizontal propagation

direction. Typically, these models utilize stability classes to parameterize the turbulent dispersion properties. Fitting a model to the image can theoretically account for plume mass below the detection limit, which is an advantage over mass-balance methods. The GPM is only valid for an ensemble of plumes due to the stochastic nature of turbulence (Varon et al., 2018; Jongaramrungruang et al., 2019). Since we use temporal averages of plume measurements, we average over short-term turbulent structures, and therefore meet the requirements for applying the GPM (figure 9). As we also observe the vertical plume

rise with our observations, we use the bent-over Gaussian plume model by Janicke and Janicke (2001) which also accounts for the plume rise driven by buoyancy and initial vertical velocity after the release (Figure 5). This allows us to simulate an hourly averaged observation of our camera from a set of parameters. The model input contains ambient conditions and source parameters, including the emission rate. We use the model to simulate an ensemble of observations and fit the simulation to the observations to estimate the emission rate. Section 4.1 describes the model for three-dimensional, bent-over Gaussian plume

shapes, and section 4.2 explains how we simulate an observation from the model. Section 4.3 describes the a priori data used for the emission estimation, and section 4.4 how we find emission estimates and uncertainties from the inversion.





## 4.1 Gaussian plume model

Since we observe the plumes in a horizontal viewing geometry, the model needs to account for the bent-over plume shape (Figure 5). We use the plume rise model of Janicke and Janicke (2001) to calculate the plume properties along the central plume travel axis. Figure 5 shows the simulated plume shape for an ensemble of initial conditions. The properties are given as a set of discrete points $P$, which contain the spatial coordinates $x, y, z$, the plume radius $R$, the distance along the plume axis $s$, and the mass concentration $c$ of a carried quantity, e.g., $CO_2$. The model requires ambient wind velocity $u_a$, temperature $T_a$, pressure $p_a$, and relative humidity $RH_a$ as well as exhaust gas initial velocity $u_e$, temperature $T_e$, and concentration $c_0$ as input parameters (see section 4.3). The total gas enhancement in the plume depends linearly on the emission rate $Q$. The concentration $c_0$ in the plume right above the chimney follows from

$$c_0 = \frac{Q}{\dot{V}}, \tag{18}$$

where $\dot{V}$ is the air volume flux from the chimney. The concentration $c_0$ holds for a homogeneous plume segment of cylindrical form and radius $R$. We transfer each plume segment from a cylindrical concentration profile to the Gaussian profile

$$c(r) = c^* \exp\left(-\frac{r^2}{b^2}\right), \tag{19}$$

where $c^*$ is the core concentration, $r$ the distance to the central axis, and $b$ the plume width. For each segment of the plume, $c^*$ follows from the conservation of mass compared to a cylindrical plume segment with concentration $c$ from

$$c \cdot \pi R^2 \Delta s = \int_{s_0}^{s_1} ds \int_0^{2\pi} d\phi \int_0^{\infty} r\, dr\, c(r) \tag{20}$$

$$= \Delta s \cdot 2\pi \cdot \frac{c^* b^2}{2} \tag{21}$$

$$\Rightarrow c^* = c \cdot \frac{R^2}{b^2}, \tag{22}$$

where $s$ is the distance along the plume axis, i.e., $\Delta s$ the plume cross-section segment thickness.

We create a three-dimensional domain around the central plume axis as our model domain, which covers at least a $4b$ radius around each point on the plume axis. The spatial resolution of the domain cells is a third of the HySpex pixel width (approximately $1\,\mathrm{m}$). For each domain grid cell in the vicinity of the central axis, we find its distance $r$ to the plume axis and the mass concentration $c(r)$ at this distance from equation (19). We introduce two more parameters to equation (19), which are later used as fitting parameters for the inversion (section 4.4). The parameters are a scaling factor $k_c$ of the concentration $c^*$ and a scaling factor $k_b$ of the plume width $b$. We introduce them in a way that the total mass of the plume depends linearly on $k_c$ and is independent of $k_b$. Thus, the concentration scaling represents the source strength, while the width scaling accounts





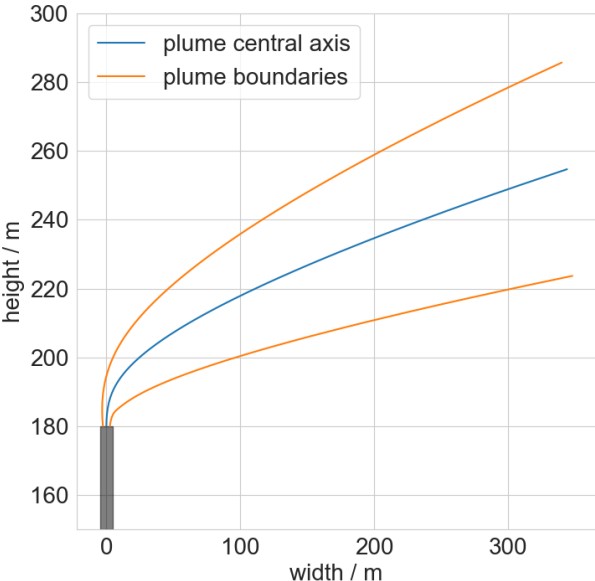

**Figure 5.** Example for the output of the Gaussian plume model IBJpluris with a chimney of $180\,\text{m}$ height. The examples input parameters are the ambient wind velocity ($7.0\,\text{m/s}$), ambient temperature ($27°\text{C}$), and relative humidity ($40\%$) at $200\,\text{m}$ height. Furthermore, the initial velocity ($13.4\,\text{m/s}$), temperature ($63.0°\text{C}$), and initial concentration ($189\,\text{g/m}^3$) of the exhaust gas are given. IBJpluris calculates the central plume axis (blue) and the plume boundaries (orange). Output parameters like the concentration of carbon dioxide are provided along the central plume axis.

for the turbulent diffusion during the time of observation and needs to be fitted (Carhart and Policastro, 1991). The total mass $M_s$ in each slice of the plume is given by

$$330 \quad M_s = \int\limits_{s_0}^{s_1} ds \int\limits_0^{2\pi} d\phi \int\limits_0^\infty r\,dr\, \frac{k_c}{(k_b)^2}\, c_s^* \exp\left(-\frac{r^2}{(k_b b_s)^2}\right) \tag{23}$$

$$= \pi \cdot \Delta s \cdot b_s^2 \cdot k_c c_s^*, \tag{24}$$

where $b_s$ and $c_s^*$ are the radial width and core concentration of the segment from $s_0$ to $s_1$, respectively. Thus, the parameter $k_b$ scales the plume width without changing its mass, while $k_c$ scales the mass and in the plume, which is linearly related to the source emission rate $Q$.





## 4.2 Observation forward model

We simulate a plume observation from the three-dimensional Gaussian model output by projecting the plume on our plane of observation and aggregating the mass according to the image pixels. A plume cell at $(x, y, z)$ is projected on the observation plane that is perpendicular to the viewing direction. Figure 6 shows a sketch which explains the projection. The observation angle $\phi$ is the angle between the plume travel direction and the viewing direction, the plume cell angle $\theta = \arctan(x/y)$ is the angle between the location of a plume parcel location and the travel direction, and the projection angle $\beta$ is the angle between the plume parcel and the observation plane. These three angles add up to $90°$. Thus, given the cell location $(x, y, z)$ and the observation angle $\phi$, the projection angle $\beta$ is given by $\beta = 90° - \phi - \theta$. The observation angle is defined between $-180°$ and $+180°$, where a negative angle denotes a plume moving to the left, a positive angle a plume moving to the right, and a zero angle a plume moving straight away from camera. The projection of the cell location $(x, y, z)$ to the observation plane $(x', z')$ is given by

$$
x' = \sqrt{x^2 + y^2} \cdot \sin(\phi + \arctan(x/y)),
$$
$$
z' = z, \tag{25}
$$

where $\phi$ is known from the measurement geometry and the ambient wind direction, while $(x, y, z)$ are the plume grid cells. We calculate a $CO_2$ mass distribution $m(x', z')$ in the observation plane by multiplying each concentration with the cell volume and projecting it according to equation (25). The simulated image is then given by aggregating the mass points $m_{jk}$ in the pixels of the observation and converting them to a column enhancement $\tilde{\alpha}_{jk}$ using

$$
\tilde{\alpha}_{jk} = m_{jk} \cdot \frac{\nu_{CO_2}}{A_{jk}}, \tag{26}
$$

where $\nu_{CO_2} \approx 0.509 \, \mathrm{m^3/kg}$ is the specific volume of $CO_2$ at normal conditions and $A_{jk}$ is the pixel area.

We choose all input parameters of the model to be constant for an observation except for the ambient wind speed (section 4.3). Thus, a simulated observation has a total of four independent parameters, which are ambient wind speed $u_a$, observation angle $\phi$, emission scaling $k_c$, and the plume width scaling $k_b$.

## 4.3 A priori data

Plume formation in the atmosphere depends on ambient meteorological conditions like temperature, pressure, humidity, and wind velocity. The exhaust gas temperature, initial velocity, and source shape and diameter also affect the plume formation. The operator of the GKM provided us with their operational data for all days on which we have been observing the GKM exhaust plume. The power plant operates four different sections, namely units 6, 7, 8, and 9. The stack height of the power plant is $180 \, \mathrm{m}$ for unit 9 and $200 \, \mathrm{m}$ for units 6 to 8. Their data contain, for each unit, the coal consumption, the exhaust volume flow rate at the chimney top, the chimney tip diameter, and the exhaust gas temperature. Unit 9 is the youngest facility in



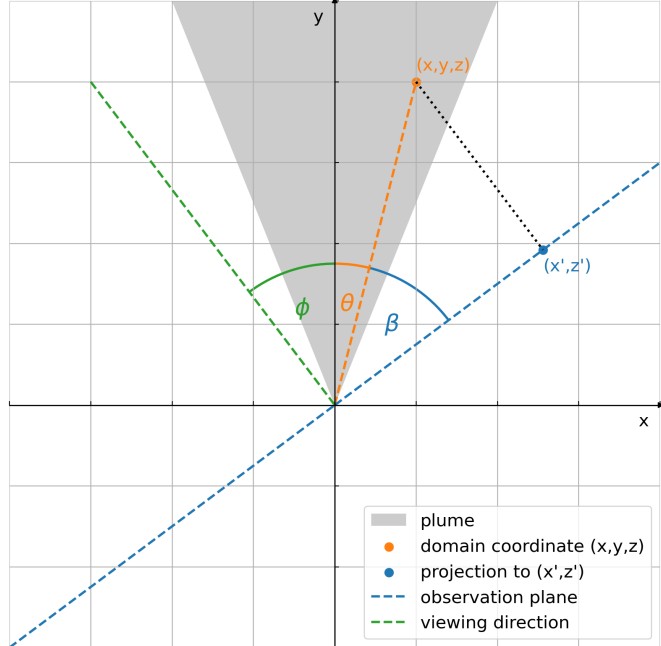

**Figure 6.** Sketch of the projection of a plume cell at $(x, y, z)$ to the observation plane $(x', z')$. The gray shading shows a conceptual top-down view on a horizontal plume cross-section - the triangular shape is chosen for simplicity and does not represent the actual plume shape. The camera viewing direction is marked by the green line, such that the angle $\phi$ is the angle between the viewing direction and the y-axis. The orange line points to an arbitrary plume cell at $(x, y, z)$, with a plume cell angle of $\theta = \arctan(x/y)$. The blue line marks the projection plane which is perpendicular to the viewing direction. Thus, the projection angle $\beta$ is given by $\beta = 90° - \phi - \theta$. Using the distance $\sqrt{x^2 + y^2}$ of the plume cell to the origin, the projected length coordinate follows from elementary geometry to $x' = \sqrt{x^2 + y^2} \cdot \sin(\phi + \arctan(x/y))$.

operation since 2015. It contributes a maximum gross power production of $911\,\mathrm{MW}$ to the power plant total of $2146\,\mathrm{MW}$. The

GKM also provided us with the instantaneous gross power production of unit 9. Furthermore, the GKM dataset includes $10\,\mathrm{m}$ meteorological data at the power plant location. This data consists of wind velocity and direction, temperature, pressure, and relative humidity. Both the meteorological and operational datasets have a resolution of $1\,\mathrm{min}$.

We calculate the expected emission rate of $CO_2$ from power production using the official German emission factor for hard coal of $93.1\,\mathrm{tCO_2/TJ}$ (Sandau et al., 2021). Unit 9 is a modern unit with an efficiency of 46.4% (Grosskraftwerk, 2015),

giving an emission factor of $722\,\mathrm{gCO_2/kWh}$. We calculate the power plant's mean emission factor from its annual reported power production and the $CO_2$ emissions reported to the European Pollutant Release and Transfer Register (E-PRTR, table S1). The total power plant has an average emission factor of $955\,\mathrm{gCO_2/kWh}$, leading to an estimated average emission factor of $1127\,\mathrm{gCO_2/kWh}$ for the older units 6, 7, and 8, for which efficiencies are unavailable. These factors allow for estimating emission rates from the instantaneous power production. Thus, the GKM operator provided us with all the necessary source

input parameters for the Gaussian plume model. On May 13, 2022, unit 9 of the GKM was malfunctioning, and the power



production was shifted to unit 6. The Fraunhofer-Institut für Solare Energiesysteme (ISE) allocates the power production of
German power plants with $15\,\mathrm{min}$ resolution. We use their data to estimate the power production of unit 6 on this day. The
ambient input parameters were partly provided by the meteorological data $(p_a, T_a, RH_a)$ of the GKM and partly by our wind
lidar $(u_a, \phi)$. We measure wind speed and direction in six height levels, the two uppermost being $150\,\mathrm{m}$ and $200\,\mathrm{m}$ above

ground level, from which we derive the a priori wind data at plume height. A $20\,\mathrm{min}$ running mean removes high-frequency
fluctuations from the lidar data due to turbulence within a laser rotation. We average all input parameters over the same period
used for the HySpex measurements, i.e., approximately 1 hour. Typical wind speeds during observation are $5\,\mathrm{m/s}$, for which
we find hourly standard deviations of $0.9\,\mathrm{m/s}$. Depending on meteorological stability during observation, the uncertainty of
the wind direction varies between $18°$ and $44°$. One exception to the a priori wind data is the observation on Sep. 8, 2021.

Since we do not have lidar data for this day, we scaled the $10\,\mathrm{m}$ winds of the GKM data to shaft height using data from the
Copernicus Atmospheric Monitoring Service (CAMS) reanalysis (Inness et al., 2019). Figure S1 shows both the scaled wind
speed and typical lidar observations. All input parameters for the Gaussian plume model are listed in table S2.

## 4.4 Inverse estimate

As described in section 4.2, there are four parameters that we can vary to fit the simulated image to the observed one. These

are the plume width scaling $k_b$, the emission scaling $k_c$, the plume velocity $u_a$, and the observation angle $\phi$. Despite wind
lidar measurements being available, we use $u_a$ and $\phi$ as free parameters since the wind lidar measurements are performed at
kilometer distance from the source, and typical plume heights are above the lidar top height of $200\,\mathrm{m}$.

We use a brute force method to scan over a space of parameter sets and compare the simulated observation with the obser-
vation. The plume mask is defined as the largest continuous patch of enhancements above twice the noise level $\sigma_j$ given in

equation (11). Figure 7a) shows an example of an observed plume from the GKM. Measurements in the same frames as the
chimney are excluded since the comparatively high brightness of the chimney affects the other spectra in the frame. Figure 7b)
shows an example of a simulated plume. The translucent area is below the $2\sigma_j$ noise level, while the colored area is above $2\sigma_j$.
Enhancements above $2\sigma_j$ noise level are well above the detection limit; thus, the matched filter retrieval should detect them.
We use the union of the plume mask and all pixels with an enhancement above $2\sigma_j$ of the noise level as our fit mask. While

computationally costly, the brute force method provides further insights into the parameter space. For each parameter set, we
calculate the reduced chi-square $\chi_r^2$ by

$$\chi_r^2 = \frac{1}{N-4} \sum_{jk}^{\mathrm{fitmask}} \left( \frac{\tilde{\alpha}_{jk}(k_b, k_c, u_a, \phi) - \alpha_{jk}}{\sigma_j} \right)^2 , \tag{27}$$

where $N$ is the number of pixels in the fit mask, $\tilde{\alpha}_{jk}(k_b, k_c, u_a, \phi)$ is the simulated column enhancement in pixel $jk$ for the
parameter set $(k_b, k_c, u_a, \phi)$, $\alpha_{jk}$ is the observed column enhancement in pixel $jk$, and $\sigma_j$ is the uncertainty of the observed

column enhancement for the pixels in line $j$. Figure 7c) shows the residual of the observation and the best fit simulation.

Since we scan the parameter space in a brute force manner, we calculate the reduced $\chi_r^2$ for each parameter set in a wide
parameter range. We use the optimal parameter set $(\hat{k}_b, \hat{k}_c, \hat{u}_a, \hat{\phi})$ with the lowest $\chi_r^2$ for our emission estimate. Figure 8 shows



an illustrative $\chi_r^2$ hypersurface of the four-dimensional parameter space along the cross-sections through the optimal parameter set for the plume on 2022/03/26 retrieved from the averaging period between 15:56 - 17:36 UTC. The $\chi_r^2$ surfaces are smooth, which indicates that $\chi_r^2$ is a continuous function of the parameters. There is a unique minimum on each hypersurface, marked by a blue dot, around which the $\chi_r^2$ increases monotonically. For purely statistical errors, an increase of 1 in $\chi_r^2$ corresponds to a mean deviation of one standard deviation ($1\sigma$) between the simulated and observed image due to parameter changes (Bevington et al., 1993). Thus, we consider all parameter sets with

$$\chi_r^2 < \min(\chi_r^2) + 1 \tag{28}$$

for the uncertainty of the emission estimate. Fixing $(\hat{k}_b, \hat{u}_a, \hat{\phi})$ to the optimal parameter set, we can estimate the uncertainty of the emission estimate by varying $k_c$ within the range of equation 28. The minimum and maximum emission estimates are used as uncertainty ranges. This estimate neglects systematic errors between observation and simulation, which are challenging to account for. Therefore, we use the criterion as our best approximation of the uncertainty range.

The shape of the well in the $\chi_r^2$ hypersurface reveals correlations between the parameters. A circular well indicates that the parameters are uncorrelated, such as for the well spanned by the wind speed parameter $u_a$ and the plume width parameter $k_b$. Flat wells suggest that the observations do not constrain the parameter strongly since the $\chi_r^2$ does not change much with the parameter. Figure 8 shows that the observation angle is the least constrained parameter and that there is a correlation between the plume width parameter $k_b$ and emission scaling factor $k_c$. This correlation is expected since the plume width scaling distributes the total mass over a larger area, while the emission scaling can increase the overall mass to match the observation again. Observations under favorable conditions constrain the ambiguity well since they provide enough information on plume width. If the observation is taken under challenging conditions, such as clouds or low emission rates, the plume width is not well constrained, which leads to a flat, slant well in the plume width ($k_b$) - emission scaling ($k_c$) plane. Some observations show a correlation between wind speed and wind direction (e.g., figure S4), which is explained by an ambiguity in the observed plume shape for horizontally viewing observers. A plume traveling perpendicular to the viewing angle at a slow wind speed will look the same as a plume traveling at a higher wind speed in a slant observation angle. In theory, the emission scaling is unaffected by this ambiguity. This becomes clear using a simple mass balance argument. The observation gives the plume mass, while the travel time of the plume is approximately given by plume length divided by wind speed. A geometric observation factor will act on both speed and direction similarly, thus canceling out in the travel time and emission estimate.

The $\chi_r^2$-surfaces alongside a plot of the best fit for each observation are given in the supplementary material. We sample the $\chi_r^2$-surfaces with step sizes of 0.2 - 0.3 m/s for the ambient wind speed, 5° for the wind direction, 0.1 for $k_c$ and 0.1 - 0.4 for $k_b$. We sample the $k_c$ dimension a final time with 0.01 step size at the optimal parameters $(\hat{k}_b, \hat{u}_a, \hat{\phi})$ to improve the emission estimate given the other three parameters.





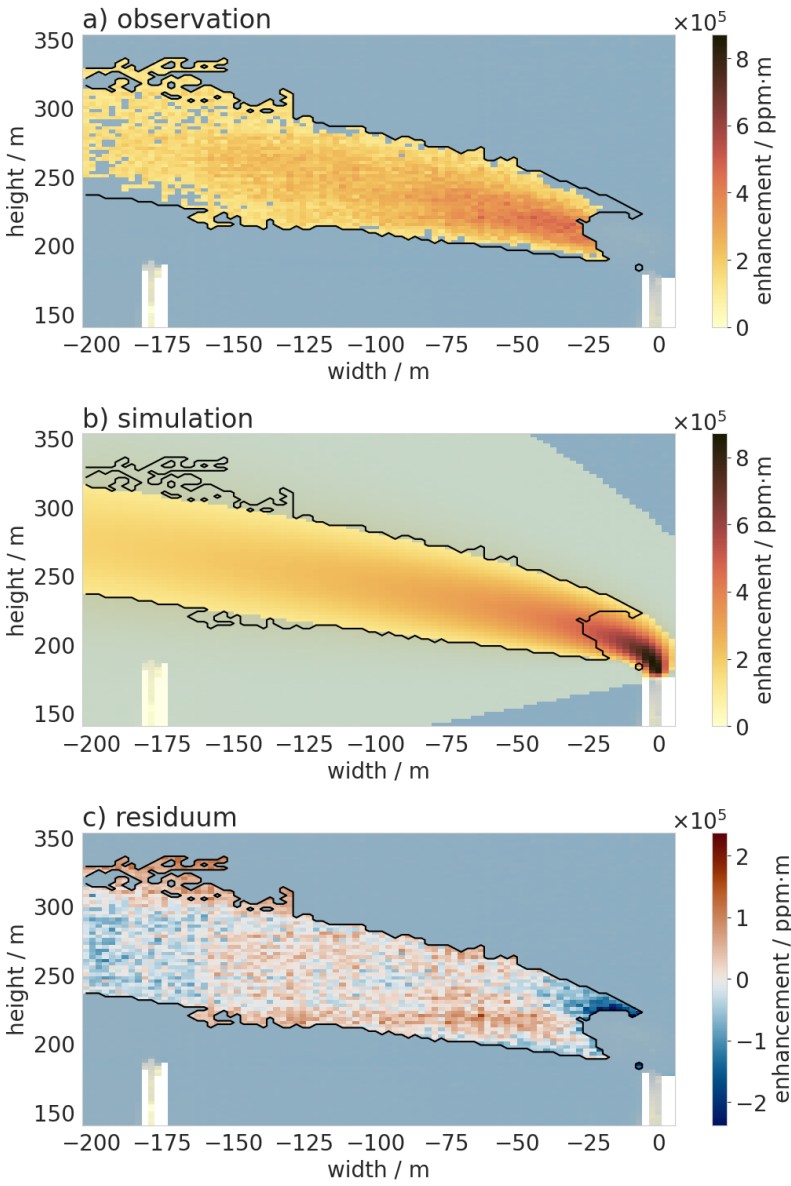

**Figure 7.** Observed (panel a) and simulated (panel b) $CO_2$ plume from the GKM Mannheim. The simulations correspond to the optimal parameter set. Residuals are shown in panel c. The black contour in each panel marks the fitmask, which is the union of the plumemask from panel a and all pixels with a simulated enhancement larger than $2 \cdot \sigma_j$ (bright colors in panel b). The part above the chimney is excluded since the chimney is so bright that it affects the retrievals for the respective entire frames. The example is the observation from Mar. 26, 2022, 15:56 - 17:36 UTC.




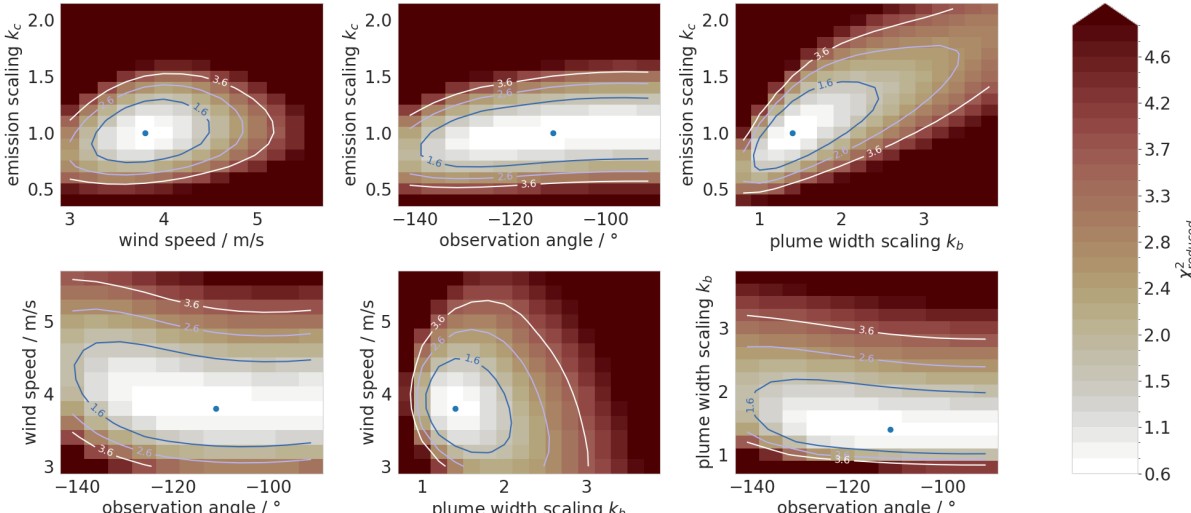

**Figure 8.** Cross-sections through the $\chi_r^2$ hypersurface (color coded) for the four-dimensional parameter space: emission scaling factor $k_c$, plume width factor $k_b$, wind speed $u_a$, observation angle $\phi$. The contour lines mark where the $\chi_r^2$ increases by 1, 2, and 3. The blue dot marks the minimal $\chi_r^2$. Note that all cross-sections involving the observation angle are symmetric around -90°, since a plume moving away from or toward us looks identical. The example is the observation from Mar. 26, 2022, 15:56 - 17:36 UTC.

## 5 Results

We present $CO_2$ plume images from the GKM for five days in 2021 and 2022. On each day, we measured hyperspectral images
for several hours, as listed in table 1. Figure 9 shows the total of 11 retrieved plumes identified by the iterative matched filter
algorithm described in section 3.2. In every observation, the plume is clearly visible and can be attributed to a power plant
chimney. For each of the plume images, we estimate the emission rates according to section 4.4.

The observations on 2021/09/08 were taken under favorable conditions with clear skies and a priori observation angles of
-53±13°. The sky, though, became more heterogeneous in the afternoon, causing the retrieval noise to increase (figure S3).
On 2022/03/23, there was significant condensation in the early stages of the plume, and a slant observation angle (33±28°)
poses further challenges for the retrieval of the plume images collected on that day. On 2022/03/26, we again had favorable
conditions allowing for measuring two plume images. The power plant ramped up its power production during our observation
period, so we can compare estimates over a range from 223 t/h to 455 t/h. A Sahara dust event increased the aerosol load on
2022/03/28, which caused the sky to be hazy and bright in the shortwave infrared. On 2022/05/13, we observed power plant
unit 6 instead of unit 9.

The plume images collected under unfavorable conditions on 2022/03/23 and 2022/05/13 are particularly well-suited to
illustrate the limitations of the method. The slant observation angles on 2022/03/23 reduce the apparent plume size due to
the unfavorable projection. Furthermore, condensation in the early stages of the plume removes a considerable part of the
plume. Thus, we consider these observations challenging for the retrieval. The plume images on 2022/05/13 are taken under





comparably high aerosol load with AOT in the range of 0.065 (at 2000 nm) and favorable wind conditions of approximately
5.5 m/s at 62° observation angle. Yet, the plume images show unexpected shapes that are not reproducible by the Gaussian
plume model (figure S2). We observe two enhancement patches, one above and below the expected plume shape. One stripe-
like patch is around 290 m height, while another patch is below the chimney. A possible reason for these patterns is spectral
artifacts due to background heterogeneity, e.g., thin clouds. Irrespective of the origin of these artifacts, the Gaussian model is

not capable of representing these patterns, and thus, our approach is ill-suited for emissions estimates from these images.

For each of the observations, we calculate the $CO_2$ emission rates as described in section 4. Since we have precise knowledge
of the power plant's power production from the company itself, we can compare our retrieved emissions to the correlative
bottom-up calculations. These expected emissions are considered errorless in comparison. Figure 10 shows how our retrieved
emissions compare to the expected ones calculated as described in section 4.3. Table 2 lists the retrieved emissions with the

optimal parameter sets for each observation.

We consider seven out of the 11 plume images to be taken under favorable conditions. For these observations on 2021/09/08,
2022/03/26, and 2022/03/28, we find an overall reasonable agreement between retrieved and expected emissions. The retrieved
emissions average to 89% of the expected emissions and have a mean relative uncertainty of 25%. Five of the seven observations
agree with the expected emissions within their uncertainties. Notably, the retrieved emissions agree well with the variability

of the expected emissions from 223 t/h to 455 t/h on 2022/03/26, indicating that the method can observe diurnal changes
in emission rates. The plume on 2022/03/28 was observed during a high aerosol load and small wind speeds ($\hat{u}_a$=1.4 m/s),
which gives a preliminary lower limit for the wind speed necessary for the method to work. Observations on 2021/09/08
between 14:24 and 16:35 UTC agree with the expected emissions within the uncertainty range, while observations between
12:13 and 14:23 UTC underestimate the expected emissions. The estimated observation angle $\hat{\phi}$ is between -35° and -30° for

these observations. This is significantly steeper than the a priori observation angle of -53±13°, which was derived from ERA5
data (section 4.3) since there is no wind lidar data on 2021/09/08. Observations on 2022/03/23 indicate that slant observation
angles may cause emission underestimation, which might also apply to the observations on 2021/09/08. Potential sources of
systematic errors in the retrievals are background heterogeneity of the scene, $CO_2$ features in the image region of the reference
spectrum, or assumptions in the unit absorption spectrum calculations like aerosol content.

Observations on 2022/03/23 have been taken under challenging conditions as described above. The measurement between
14:51 and 16:13 UTC underestimates the expected emissions significantly by 60% to 70%. We find a retrieved observation
angle $\hat{\phi}$ of 15° for this period, which agrees reasonably well with the a priori observation angle of 33±30°. The measurement
from 15:35 to 17:36 UTC agrees with the expected emissions within the uncertainty, but its observation angle $\hat{\phi}$ is 65° is
inconsistent with the lidar observation of 33±25°. Thus, slant observation angles and plume condensation are considered

factors that limit the method's applicability and may be used as filter criteria. The observations on 2022/05/13 overestimate
the power plant emissions. The enhancements outside the expected plume cause an increase in the width scaling factor in
the GPM inversion since the simulated enhancements need to spread out (figure S22). While the width scaling factors range
typically between 1.2 and 2.0, we find width scaling factors of 7.0 for these observations. The additional mass is attributed
to the power plant emissions, leading to overestimation of the emission scaling factor. For the observation on 2022/05/13,





**Figure 9.** Plume images for the periods listed in 2. The yellow to red colorbar is shared over all plumes and shows the atmospheric $CO_2$ enhancement in ppm·m. The blue (dark) to white (bright) color in the background shows the spectrum saturation. The bright rectangular shapes in the lower part of the images are the power plant chimneys.





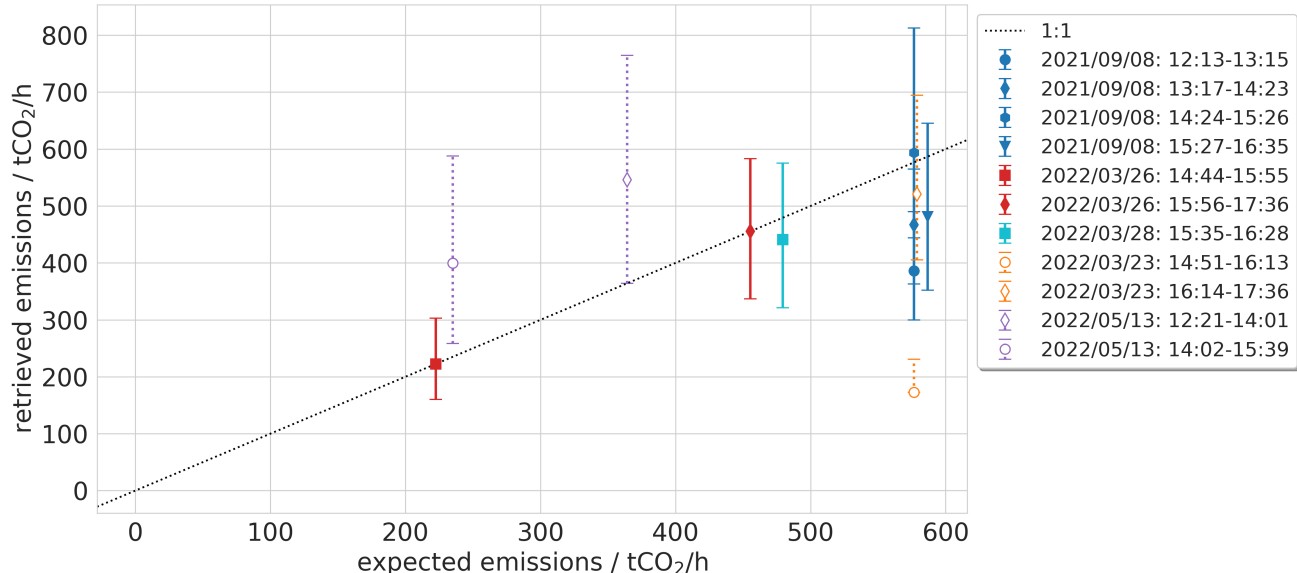

**Figure 10.** Correlation of retrieved emissions and expected emissions. Every color represents a different observation day, while each symbol represents a different time interval. Filled symbols mark observations under favorable conditions, while open symbols mark observations under unfavorable conditions. The black dashed line marks the 1:1 line.

such biased observations can be identified by non-compliance with a Gaussian plume shape and excluded from the emission estimate. However, in other cases, the effect might be too small to be identified visually but still large enough to propagate in the emission estimate.

## 6   Conclusions

We report on a proof of concept for estimating $CO_2$ emission rates for a coal-fired power plant using a ground-based hyperspec-
tral camera. The power plant is located in Mannheim, Germany, and has yearly $CO_2$ emissions of more than $4.9\,\mathrm{MtCO_2/yr}$. We demonstrate our capability to reliably image $CO_2$ plumes from individual chimneys with 11 observations over five days in 2021 and 2022. The camera observes sky-scattered sunlight in the shortwave infrared spectral range above the chimneys of the power plant. We use an iterative matched filter algorithm to retrieve the $CO_2$ enhancements from the observed spectra. Scattering on molecules and aerosols is inefficient in the infrared spectral range. Thus, the observed signal is small, and we
need to average over more than 50 minutes to reach a sufficient signal-to-noise ratio in typical cases. Averaging over such a period typically causes the observed plume to be of Gaussian shape in good approximation. Therefore, we estimate emissions by fitting a Gaussian plume model to the observed plume. The forward model is based on the plume rise model by Janicke and Janicke (2001). We estimate the ambient wind velocity and direction, the plume width, and the source emissions by minimizing





**Table 2.** Column one lists the observation time period, and column two the expected emissions $E_{exp}$ (section 4.3). The retrieved emissions $E_{ret}$ (section 4.4) are listed in column three, with the uncertainty range in parentheses. Columns four to seven give the optimal inversion parameters $\hat{k}_c$, $\hat{k}_b$, $\hat{\phi}$, and $\hat{u}_a$. Note that $\hat{k}_c$ is the relative scaling between retrieved and expected emissions i.e. it represents the relative deviation. The last column shows the minimum $\chi_r^2$ for each observation.

| Date and time | $E_{exp}$ [t/h] | $E_{ret}$ [t/h] | $\hat{k}_c$ | $\hat{k}_b$ | $\hat{\phi}$ | $\hat{u}_a$ | minimal $\chi_r^2$ |
|---|---|---|---|---|---|---|---|
| 2021/09/08: 12:13 - 13:15 | 576 | 386 (300 - 490) | 0.67 (0.52 - 0.85) | 1.60 | -34.00 | 8.30 | 1.85 |
| 2021/09/08: 13:17 - 14:23 | 576 | 467 (363 - 565) | 0.81 (0.63 - 0.98) | 1.80 | -30.00 | 10.10 | 3.25 |
| 2021/09/08: 14:24 - 15:26 | 576 | 594 (444 - 813) | 1.03 (0.77 - 1.41) | 1.80 | -35.00 | 7.60 | 0.66 |
| 2021/09/08: 15:27 - 16:35 | 587 | 481 (352 - 645) | 0.82 (0.60 - 1.10) | 1.75 | -25.00 | 9.20 | 0.65 |
| 2022/03/26: 14:44 - 15:55 | 223 | 223 (160 - 303) | 1.00 (0.72 - 1.36) | 2.00 | -141.00 | 3.60 | 0.29 |
| 2022/03/26: 15:56 - 17:36 | 455 | 455 (337 - 583) | 1.00 (0.74 - 1.28) | 1.40 | -111.00 | 3.80 | 0.63 |
| 2022/03/28: 15:35 - 16:28 | 479 | 441 (321 - 575) | 0.92 (0.67 - 1.20) | 2.00 | -65.00 | 1.40 | 0.85 |
| 2022/03/23: 14:51 - 16:13 | 576 | 173 (173 - 231) | 0.30 (0.30 - 0.40) | 1.20 | 15.00 | 5.70 | 1.43 |
| 2022/03/23: 16:14 - 17:36 | 579 | 521 (405 - 695) | 0.90 (0.70 - 1.20) | 1.80 | 65.00 | 3.90 | 1.24 |
| 2022/05/13: 12:21 - 14:01 | 364 | 546 (364 - 765) | 1.50 (1.00 - 2.10) | 7.00 | 40.00 | 6.20 | 1.77 |
| 2022/05/13: 14:02 - 15:39 | 235 | 400 (259 - 588) | 1.70 (1.10 - 2.50) | 7.00 | 80.00 | 6.60 | 3.00 |

the $\chi_r^2$ between the observed and simulated plume. Therefore, we sample the $\chi_r^2$-space using a brute force approach, which,

for all cases, reveals an unambiguous minimum in $\chi_r^2$. For validation, we calculate the expected emissions based on the power plant's power production during the observation.

Favorable observation conditions are homogeneous skies, stable wind speeds, and a wind direction perpendicular to the viewing direction. We present seven observations taken on three days with ambient conditions matching such favorable conditions. For these seven observations, the estimated emission rates average to 89% of the expected emissions with a mean relative

uncertainty of 25%; thus, they agree reasonably well with the expected emissions. Observations indicate that we can follow the diurnal trend of the power plant emissions under such conditions and that our technique works in wind speeds down to ~1.4 m/s.

We also present plumes that show enhancement artifacts, plume condensation, and unfavorable wind conditions to demonstrate the limitations of the technique. Obvious non-compliance with the Gaussian plume shape causes inaccurate emission

estimates, but these cases are easily identified in the image and neglected for further analysis. Furthermore, we find that plume condensation and steep observation angles pose challenges for the technique. More observations are needed to quantify the impact of these effects on the emission estimates and to develop quantitative and suitable filter criteria when measuring under non-favorable conditions.

Our spectral imaging technique adds to the pool of tools to verify $CO_2$ emission rates of localized sources. In that context, our

ground-based setup allows for monitoring individual sources over prolonged periods, which, for example, is complementary to the snap-shot images provided by satellites. We envision that further development of our technique can provide independent



data for emission inventories and can be used to verify bottom-up emission estimates. Our instruments fit in a car, and the ground-based observation geometry enables us to choose the targets flexibly. Potential future targets are less well-known anthropogenic sources such as facilities in the chemical, metallurgy, or cement industry, and natural $CO_2$ sources like volcanoes.

*Data availability.* Hyperspectral data is available on reasonable request. Operational data of the Grosskraftwerk Mannheim is available on request with consent of the power plant operator.

*Author contributions.* MK performed the observations, adapted the retrieval algorithm, developed plume model and inversion scheme, and wrote the manuscript. RK helped with the instrumental characterization and deployment. SV gave advice on the plume model and inversion scheme. FK, HH, MS, and TS helped with the observations and the retrieval algorithm. MS implemented the AHRS system. BB provided
data of the Fraunhofer-Institut für Solare Energiesysteme (ISE). AB developed the research question, supervised the project, and helped with the manuscript. All authors have read and contributed to the manuscript.

*Competing interests.* At least one of the (co-)authors is a member of the editorial board of Atmospheric Measurement Techniques.

*Acknowledgements.* The research presented here has been funded by the Deutsche Forschungsgemeinschaft (DFG, German Research Foundation) - Project numbers 414210689, 449857152. We thank Roland Krupp of the GKM Mannheim for providing us with high quality data
and information on their power plant operation. We thank Bernhard Vogel and Frank Wagner for their effort in establishing and maintaining AERONET site in Karlsruhe. We thank Tobias Schmitt and Benedikt Löw for their advice on retrieval algorithm and inversion scheme.



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
