# Peer review of "Quantitative imaging of carbon dioxide plumes using a ground-based shortwave infrared spectral camera"

_EGUsphere, 2023_

## Author Comment (AC1)

GENERAL COMMENTS

Estimating the emission amounts is more challenging than detecting the enhancement due to large uncertainty in wind speed and direction. Retrieving plume heights, wind height and direction is difficult from a snapshot of satellite data. The measurement technique and analytical method described here are unique. The authors are measuring the plume structure with an imaging spectrometer and a wind lidar and then comparing with a model. The paper demonstrates the difficulty and importance of field measurements. 5-day data set with various weather condition are very valuable for scientific community. Minor revisions will help readers' understanding. I recommend publication after revisions.

Thank you very much for the positive judgement of our work and the helpful comments.

SPECIFIC COMMENTS

(1) Page 2, Line 33, analogue techniques

An additional explanation on "analogue techniques" is helpful.

The techniques for estimating methane and carbon dioxide emissions from satellite and airborne imagery are identical in most respects, although there may be differences in detail, e.g. the use of nitrogen dioxide as a plume mask proxy for carbon dioxide. As the paper only deals with carbon dioxide, we have removed the part "using analogue techniques".

(2) Page 4, Figure 1

Descriptions such as the height of the chimney, distance between three chimneys, and distance between spectrometers and the chimneys in the figure or the figure caption will helpful.

We added the sentence

"The observed chimneys are 200 m (left) and 180 m (right) high and are approximately 3.2 km away from the camera."

to the caption. Exact distances between camera and the observed chimney are listed in table 1.

(3) Page 8, Line 185, "sparsity constraint on enhancement"

A brief description will help readers without referring the paper.

We added the sentence

"The sparsity constraint sets enhancements below the detection limit to zero, which enables the matched filter to iteratively remove the target signal from the background clutter estimation."

To line 187 of the manuscript.

(4) Page 25, Line 507 "Favorable observation condition"

Disadvantage of this observation is a weak scattered light as a light source. Scattering depends on the geometry of the sun, target, and observer. Is the back-scattered geometry such as "the sun is behind the observer" favorable?

Since Rayleigh scattering is inefficient in the SWIR spectral range, the largest contribution of light comes from scattering at aerosols. The Henyey-Greenstein phase function for aerosol scattering is asymmetric and favors forward scattering, with typical asymmetry parameters above 0.5 [1], between 0.76 and 0.82 at the measurement days according to AERONET (table 1 in manuscript). Therefore, the back-scattered geometry is not advantageous with respect to the illumination. Yet, the sun should be far away from the field of view to avoid strong gradient inside the image. Thus, a sun behind or above the observer is favorable.
For the same reasons cameras operating in the UV/VIS range also prefer the sun in the back, while they also profit from an increased illumination due to Rayleigh-backscattering.

(5) Supplement P2, Figure 2 the right panel

Do colors in the right panel mean something?

We believe you mean Figure S1, the wind lidar data. No, the colors have no meaning, we removed them to avoid confusion.

TECHNICAL CORRECTIONS

No specific comments.

[1] Pandolfi, M. *et al.* A European aerosol phenomenology – 6: scattering properties of atmospheric aerosol particles from 28 ACTRIS sites. *Atmos. Chem. Phys.* **18**, 7877–7911 (2018), https://doi.org/10.5194/acp-18-7877-2018.

---

## Author Comment (AC2)

The authors present a very novel ground-based observational study to estimate power plant CO2 emissions. Passive plume-mapping hyperspectral instruments have been deployed from many suborbital and orbital platforms, but utilizing a ground-based spectrometer for this plume-mapping use-case is quite new and exciting. The manuscript itself is very clear and all steps from observation to emission quantification are clearly described. The manuscript should proceed to publication, though I have a few minor comments, which I detail below.

Thank you very much for the positive judgement of our work and the helpful comments.

1. Is the dimension of your covariance matrix sufficient to constrain (i.e., not underestimate) CO2 concentration in the CMF algorithm? Reading from the text, it appears that the dimension of your data cube is 286 x 384 x 288 - with 286 being the number of frames. How many active bands do you use in your retrieval? If it's 6-7nm spectral sampling between 1900-2100, that would roughly 30 bands, so a 30x30 dimension covariance matrix. Are 286 elements sufficient? Previous studies have found that too few pixels (aerial hyperspectral imagers) in the along-track direction can result in concentration enhancements that are biased low.

We appreciate your feedback. In airborne observations, it is customary to employ a matched filter independently for each along-track line, primarily to mitigate the issue of striping. This practice is feasible due to the large number of spectra contained within each along-track observation, as small datasets have been identified as problematic. Consequently, we utilize both spatial dimensions for the matched filter estimation, thus the covariance matrix is based on approximately 100 000 pixels (350 along-track points by 286 spatial points). This matrix has dimensions of 53x53 (with a spectral sampling distance of 5.4 nm). To address the striping problem, we employ a differential approach in our matched filter. Using a line-specific reference spectrum mitigates many issues leading to striping (e.g., detector non-uniformity), which has notably enhanced our results.

This information can be found in lines 193-194 of the manuscript, and we have included an explicit formula to enhance clarity.

2. How much of the uncertainty comes from your fit mask? An attractiveness of your approach is that you only need a statistically representative sample of pixels within the plume to make an assumption about emission rates. Why consider the mask from the simulation? Looking through the plume masks in the main manuscript and the SI, seems like many of the masks incorporate areas of null enhancement within the observation. Is this biasing any of your results? Is there much of a difference if you use an observation only plume mask?

We incorporate the plume mask derived from the simulation process to account for null-enhancements adjacent to the observed plume mask. These null enhancements

provide valuable information by indicating the absence of detectable carbon dioxide in the corresponding pixel. In earlier stages of method development, we exclusively utilized the observation-only mask. However, we discovered that this approach leads to an undesired heightened correlation between the scaling of plume width and emission estimation. This effect occurs because the fitting procedure allows for the expansion of the plume width, effectively relocating emitted mass outside the designated plume mask. This removal of mass is compensated for by an increase in emission strength. Figure A illustrates an extreme case to highlight this issue, demonstrating that the resulting emission rates derived from these parameters are unrealistic and significantly differ from what we would have observed. Consequently, the inclusion of null-enhanced pixels is crucial to our analysis.

We added the sentence:

"Including background pixels in the fit mask keeps the fit close to the observation even outside the observed plume mask."

to line 401 of the manuscript.

[Figure]

*Figure A: Effect of only using the observation plume mask (red contour). The colored pixels (black to yellow) are simulated enhancements above the detection limit, gray pixels are below the detection limit. The left plot shows the best fitting scenario, while the right shows a simulation which is considered similarly good if only observation plume pixels are considered in the fit. The width scaling factor can be increased drastically, if the emission scaling factor compensates for the mass inside the observation plume mask, while the mass outside the mask increases above the detection limit. The title above the panels states the fit chi-squared value (X2), width scaling factor $k_b$ (width), and emission scaling factor $k_c$ (emissions).*

3. Figure S22 - you speak in the manuscript of the inability to get a good emission estimate on 2022-05-13 and point to problems with the width scaling factor. Curious however if you think another quantification approach could be well suited for this problem - many plume-mapping aerial/satellite algorithms use the integrated mass

enhancement method, which is mostly concerned with getting the mass of the plume correct and less the transport, rise, etc. Could this be an option for this problem, or not given the nature of the observation?

In theory, these mass balance methods are applicable to the problem. They have some caveats though.

In contrast to the Gaussian plume model, mass balance methods cannot account for enhancements below the detection limit of the method. Thus, results from such techniques are prone to be biased low, especially with a high detection limit.

A practical reason to use a Gaussian model in our observation geometry is that we observe the vertical plume rise in the initial stages of the plume. Most mass balance methods either need the plume lifetime (Integrated mass enhancement) or the flow velocity (Cross-sectional flux). In airborne applications, usually the plume velocity is taken as the wind velocity, and the lifetime is derived from the length of the plume. The cross-section is taken perpendicular to the plume travel direction, which is easily visible in a top-down measurement.
For our observations, the cross-sections cross the plume at a slanted angle, which depends on the observation angle (Figure B). To properly find the correct cross-sections and associated plume lifetimes and velocities, a plume model needs to be employed and fitted to the observation. This requires similar efforts to the full Gaussian inversion scheme presented, taking away the simplicity of mass balance approaches.

To answer your question concerning the data on 2022/05/13, we believe you might be correct. The background artifacts could be excluded by visual inspection, and a mass balance method applied to the remaining plume, as has been done in many studies. In general, however, we believe that, given our complicated side-view of the plume, the Gaussian modelling approach is superior.

[Figure]

*Figure B: The cross-sections used for mass-balance methods (blue) lie at an angle in the observed plume (white to red). The cross-section angles depend on (a) the relative angle between viewing direction and the plume travel direction, (b) the external wind field, and (c) plume parameters (e.g., thermal rise). The presented plume was created using an early version of the retrieval method and is for demonstration purposes only.*
*The cross-sections are calculated using a simple inversion scheme. For a set of wind velocities and directions, the main plume axis is found by the plume rise model of [1]. At equidistant distances along the axis (dotted lines), a Gaussian fit to the observational data along the cross-section is performed. The best matching wind velocity and direction provides the lifetime and plume flow at the cross-sections for mass balance emission estimation.*

[1] Janicke, U. & Janicke, L. A three-dimensional plume rise model for dry and wet plumes. *Atmospheric Environment* **35**, 877–890 (2001).